# Comparison of Black Tea Waste and Legume Roughages: Methane Mitigation and Rumen Fermentation Parameters

**DOI:** 10.3390/metabo13060731

**Published:** 2023-06-07

**Authors:** Gurkan Sezmis, Adem Kaya, Hatice Kaya, Muhlis Macit, Kadir Erten, Valiollah Palangi, Maximilian Lackner

**Affiliations:** 1Department of Animal Science, Faculty of Agriculture, Yozgat Bozok University, 66200 Yozgat, Türkiye; 2Department of Animal Science, Agricultural Faculty, Ataturk University, 25240 Erzurum, Türkiye; 3Department of Animal Science, Tekirdag Namik Kemal University, 59030 Tekirdag, Türkiye; 4Department of Animal Science, Faculty of Agriculture, Ege University, 35100 Izmir, Türkiye; 5Department of Industrial Engineering, University of Applied Sciences Technikum Wien, Hoechstaedtplatz 6, 1200 Vienna, Austria

**Keywords:** black tea waste, in vitro gas production, legume roughages, methane emission

## Abstract

The chemical composition, in vitro total gas and CH_4_ production and performance of cattle fed on factory black tea waste (*Camellia sinensis*) (BTW), alfalfa (*Medicago Sativa*), sainfoin (*Onobrychis sativa*) and white clover (*Trifolium repens*) was investigated. The gas production was quantified at the 24th hour of the incubation process. BTW was found to vary from roughages in chemical composition (*p* < 0.05). In addition, the roughages differed in terms of nutrient composition and gas production (*p* < 0.05). In legume roughages, acetic acid (AA), propionic acid (PA), butyric acid (BA), and total volatile fatty acids (TVFA) values ranged from 52.36–57.00 mmol/L, 13.46–17.20 mmol/L, 9.79–12.43 mmol/L, and 79.71–89.05 mmol/L, respectively. In comparison with black tea waste, legume roughages had higher values of AA, PA, BA, and TVFA. Black tea waste contained a higher acetic acid ratio than legume roughages when compared as a percentage. There was a similar ratio of propionic acid to the rate calculated for sainfoin (*Onobrychis sativa*) and clover (*Trifolium repens*), and a similar ratio of butyric acid to the ratio determined for alfalfa (*Medicago Sativa*). The current study shows that the 5.7–6.3% tannin content of black tea waste can be used in ruminant rations with high-quality roughages. Due to the fact that BTW reduces methane emissions from ruminants and eliminates energy waste from them, the environment can be improved. To obtain more reliable results, further animal feeding experiments on legume roughages and BTW are required.

## 1. Introduction

Ruminant animals need roughage due to the anatomical structure and physiological functions of their digestive systems. Roughage quality and the amount of roughage converted into yield are important factors for the animals. Those farms with insufficient or poor quality roughage sources require more concentrated feed sources for milk production, which increases ration costs and decreases profits. Consequently, dairy and beef cattle enterprises need affordable quality roughage sources. Several countries use alternative feed sources to supplement ruminant animals’ nutrition to compensate for the lack of quality roughage.

It has also been proposed that BTW can be used to provide nutrients and bioactive compounds for the feeding of ruminants, which reduces waste and feed costs while maintaining a healthy environment [1,2]. Accordingly, the use of tea waste can be economically beneficial, and the tannin content should be considered [3]. Recently, one of the roughage sources that has attracted attention as an alternative has been factory black tea waste (BTW) [4,5]. After fresh tea is processed into products, factory black tea waste forms as a by-product, causing significant environmental pollution, including ecological and economic damage, since it is not used in any other way; It does not always end up as compost, but is also thrown into nature in an uncontrolled way [6], where methane can be formed. It is reported that this waste, which is a by-product of the food industry, contains 5.7–6.3% tannin, 2442 kcal ME/kg DM (ME = metabolic energy and DM = dry mass) energy, 19.8% crude fiber, and 18.2% crude protein (dry matter basis) [7]. Due to the content of tannins in black tea waste, a study conducted on black tea waste concluded that it reduces methane emissions and avoids ruminant energy consumption [8]. It is reported that 2–12% of the total energy taken with feed, and even 2–15% in some sources, is lost in the form of methane gas energy [9]. Feeds are fermented into gases such as volatile fatty acids (VFA), CO_2,_ and methane by rumen bacteria under rumen or in vitro rumen conditions. Rumen gas is produced primarily by the fermentation of carbohydrates into AA, PA, and butyric acids [10].

As part of ruminant nutrition, secondary plant compounds, such as tannins derived from tea leaves, can have a significant impact. Due to the chemical composition and concentration of tannins, there have been various studies with sometimes contradictory results regarding the performance of ruminants when tannins are co-fed. Among the many effects of tannins (high-molecular-weight polyphenols soluble in water) on ruminal fermentation, they decrease ruminal protein breakdown, reduce methane production, prevent bloating, and increase conjugated linoleic acid concentration in feeds derived from phenolic compounds [11,12]. Despite this, many studies have found that tannins may negatively affect the digestibility of diets. As tannins are interpreted differently depending on their type, origin, and supplement levels, their influence on ruminal protozoa, bacteria, fungi, and methanogens varies [13]. According to Petlum et al. [14], molecular weight greatly influences the effectiveness of condensed tannins as ruminal methane inhibitors. There has also been a correlation between tannin potency and dose. According to Mehansho et al. [15], hydrolyzable tannins are known to be toxic to ruminants as opposed to being digestion inhibitors.

Climate change has been exacerbated by greenhouse gas emissions produced by ruminants, including the greenhouse gases (GHGs) methane (CH_4_), nitrous oxide, and carbon dioxide. The waste of energy caused by ruminant methane emissions is linked to global warming as well [16]. In addition to reducing feed energy loss and greenhouse gas emissions, animal nutritionists strive to maintain animal health, productivity, and performance. Therefore, ruminant production can be enhanced, as well as emissions of GHGs reduced, through reducing enteric CH_4_ production. Hydrogen production efficiency in the rumen is different due to the end products of the microbial fermentation of nutrients. The rumen releases hydrogen through the production of acetate and butyrate, while propionate can be used as a competitive pathway. Ruminant methane mitigation strategies must consider a number of factors, including the metabolism of hydrogen and methanogens. The tannins found in most plants’ skin, leaves, and roots are water-soluble phenolic compounds that form soluble or insoluble complexes with proteins. Tannins are chemically diverse groups of phenolic compounds. In addition to decreasing ruminal protein degradation and methanogenesis, tannins also prevent unsaturated fatty acids from being biohydrogenated in the rumen [17].

Using in vitro gas production (GP), the aim of this study was to evaluate BTW’s potential use as a replacement for legume hays, such as alfalfa (*Medicago Sativa*), sainfoin (*Onobrychis sativa*), and clover (*Trifolium repens*). This can be achieved by analyzing and comparing in vitro total gas (GP) and methane (MG) productions, performance parameters, microbial protein production values (MP), and volatile fatty acids (VFA) of BTW and legume roughages.

## 2. Materials and Methods

### 2.1. Material

The alfalfa, sainfoin (1/10 flowering), and clover (general flowering) used in the present study were collected in Türkiye, from Atatürk University Plant Production Research and Application Center, and factory waste from black tea factories was also provided (Cumhuriyet, Gündoğdu, Pazarköy, Veliköy, and Camidağı) in the central district of Rize (at 39–56 north latitude and 32–51 east longitude) as a mixture of black tea waste that processes the fresh tea plants harvested in the second and third exile periods. To determine the parameters of in vitro GP, rumen liquor was collected from 7-year-old female Brown Swiss cattle, which completed its rumen development (as soon as the animals were humanely euthanized). In addition to being purebreds, each cow was registered with its respective breed association. Twice daily, at 8:00 a.m. and 04:00 p.m., cows were fed a mixture of rations, mostly corn silage. In order to maintain relatively stable rumen conditions, it was then placed in a screw-top glass bottle and transported to the lab in a capped thermos container containing water at about 39 °C. An anaerobic environment under CO_2_ was used to filter rumen liquor for use in GP [16].

### 2.2. Method

In this study, alfalfa, sainfoin, white clover, and factory black tea wastes were dried at 65 °C, ground, and sieved at 1 mm with a 1 mm cutoff size. Dry and ground feed samples were chemically analyzed (DM, CP (976.05), EE (954.02), and crude ash (942.05) in accordance with AOAC (Association of Official Agricultural Chemists) guidelines [18]. NDF, ADF, and ADL are measured by the same methods described by Van Soest et al. [19] (NDF: neutral detergent fiber, ADF: acid detergent fiber, and ADL: acid detergent lignin).

In this study, fermentation parameters were determined from feed samples utilizing a technique described by Menke and Steingass [20]. Rumen fluid mixed with buffer solution (10 mL rumen fluid + 20 mL buffer solution) was prepared according to Menke et al. [21]. A milled feed sample (0.2 g) was incubated in 100 mL calibrated syringes containing rumen liquor for 3 replicates (3 for each incubation time with 3 blanks) to measure gas production. A correction factor of 49.61 mL per 0.200 g dry matter was applied to standard measurement conditions (University of Hohenheim). After reading the total amount of gas obtained for 24-h fermentation in the GP technique, accumulated gas was transferred to the infrared methane analyzer (Sensor Europa GmbH, Erkrath, Germany model), and the methane amount was determined as a percentage of the total gas [22] (Methane with a purity of 100% was used for calibration). Using Menke and Steingass’ equations [20], performance parameters (ME, NE_L_, and OMD) were calculated for feed samples.
ME (MJ/kg DM) = 2.20 + 0.136 × GP + 0.057 × CP + 0.0029 × CP^2^
OMD (g/100 g DM) = 14.88 + 0.889 × GP + 0.45 × CP + 0.0651 × XA
NE_L_ (MJ/kg DM) = 0.54 + 0.096 × GP + 0.0038 × CP + 0.000173 × CF^2^
where XA is ash (g/100 g DM), GP is the net gas production (mL) at 24 h, CP is crude protein, CF is crude fat, OMD is organic matter digestibility, ME is metabolic energy, and NE_L_ is net energy lactation. RFV index was calculated using the Rohweder et al. [23] formula after determining the DM intake, digestibility, and relative feed value (RFV) of samples. 

DDMI or RFV (g/kgW^0.75^) is a function of (DDM × DMI)/100.

DDM (%) is inversely related to ADF concentration, i.e., DDM decreases as ADF increases.

DMI (g/kgW^0.75^) is inversely related to NDF concentration, i.e., DMI decreases as NDF increases.

The methods reported by Blümmel and Ørskov [24], Makkar et al. [25,26,27], and Van Soest and Robertson [28] were applied for the calculations of the amount of protein biomass from rumen bacteria, and the true digestible organic matter values of BTW and legume roughage were determined. In order to compute the TOMD values of the samples, the amount of sample weighed into the glass syringe was corrected according to OM [29].
$$\mathrm{TOMD}\;=100-\left[\frac{D5-D2}{D8-D7}\times 100\right]$$

*D*2: bag weight after drying in an oven under 65 °C temperature; *D*5: ash weight after 4 h in an electric furnace with a temperature of 550 °C; *D*8: DM—Ash; and *D*7: DM of samples.

By gas chromatography, total volatile fatty acids (TVFA), acetic, propionic, and butyric acids in the rumen fluid remaining in the syringes were determined following the method by Wiedmeier et al. [30]. The following conditions were used for the GC-FID: initial temperature 50 °C, hold 2.0 min, and ramp rate 20 °C min^−1^, and final temperature 280 °C, hold 0.5 min, and ramp rate 20 °C min^−1^. At 240 °C for injection and 300 °C for the detector, the temperatures are similar. A split-less injection of 2 L was performed. In the run, N_2_ was supplied at a rate of 2 mL min^−1^. Flame ionization detectors were charged with nitrogen (30 milliliters per minute), hydrogen (44 milliliters per minute), and dry air (400 milliliters per minute). 

An evaluation of tannins was completed by adding 6.25 mL of butanol-HCl reagent to 0.01 g of samples (95 mL butanol + 5 mL HCl + 0.05 g Fe_2_SO_4_.7H_2_O). A boiling water bath (100 °C) was used for an hour; the tubes were removed, cooled, and centrifuged at 3000× *g* for 100 min. The absorbance at 550 nm was measured with a CE 2030 spectrophotometer (Cecil Instruments, United Kingdom) after the supernatant had been decanted into vials. A blank sample containing the reagent was included only in the measurements [25,26]. Microbial protein production values (MP) were calculated according to Makkar et al. [27] 

### 2.3. Statistical Analysis

The SPSS 17.0 software package was used to analyze the data, and for the purpose of determining the difference between the means, this study used Duncan’s multiple comparison test [31].

## 3. Results

### 3.1. Chemical Composition

Table 1 shows the average chemical composition of the feeds.

Significant differences were found between BTW and legume roughage in terms of examined nutrients, except for dry matter content (*p* < 0.05). Legume roughages also showed differences among themselves with regard to chemical compounds (*p* < 0.05). In addition, the CT content of BTW and VFA (mg/L) contents were found to be 3.02 and 65.29, respectively.

### 3.2. In Vitro Gas (mL) and Methane Production (mL, %)

The amounts of gas and CH_4_ produced by feeds are summarized in Table 2.

The measured differences among the legume hays and BTW, which had lower gas and CH_4_ emissions values, were significant (*p* < 0.01). According to the ranking for gas and CH_4_ emissions, sainfoin = clover > alfalfa > BTW. The observation that the BTW used in this study produced less gas than the legume roughages can be explained by the fact that it provides less useful carbohydrates for microorganisms.

### 3.3. MP, OMD, ME, and NEL Levels of Feeds

The values calculated for the MP and performance parameters of feeds are shown in Table 3 below.

In terms of MP values, the feeds were arranged as clover > sainfoin = clover = BTW. Microbial protein production was higher in feeds with high energy and protein content. Elevated microbial protein production was detected in legume roughage plants, which have high energy and protein content and low cell wall components.

Concerning the parameters ME, OMD, and NEL, the ranking is sainfoin ≥ clover = alfalfa > black tea waste (*p* < 0.01). The reason for the high metabolic energy value in sainfoin and clover roughage is found in the amount of in vitro gas released as a result of fermentation. This is because the crude protein content was greater in these legumes, because the metabolic energy of the legume feeds was calculated by considering the 24-h gas production values, and crude protein contents were greater than those of BTW.

With respect to IVTOMD values (in vitro true organic matter digestibility), the order was determined as alfalfa > sainfoin = clover grass > black tea waste. The difference between BTW and legume roughage crops was significant (*p* < 0.05). Increases in the GP value at the 24th hour of fermentation and crude protein content of the feeds raised the IVTOMD.

### 3.4. DMD, DMI, and RFV Results of Feeds

In Table 4, calculated values for the DMD, DMI, and RFV parameters of the feeds are presented.

Though RFV was developed to control the quality of Alfalfa in ABD, it is still in use for all roughages. Alfalfa is widely preferred by buyers because of its cost-effective production. Forage value is not only determined by RFV, but also by NDF and ADF based on alfalfa hay at full bloom quality. In order to calculate the RFV index, alfalfa hay is analyzed at full bloom for its forage quality.

### 3.5. The Effect of Feeds on Rumen Volatile Fatty Acid Levels

The average VFA levels formed by microbial fermentation in the rumen liquor in the GP method were determined and are shown in Table 5.

The different values among the feeds were found to be significant regarding the amounts of AA, PA, BA, and TVFA (*p* < 0.01). Leguminous roughage had higher AA, PA, BA, and TVFA (mmol/L) than those of BTW. However, when compared as a percentage, the acetic acid ratio for black tea waste was found to be higher than the acetic acid ratio calculated for legume fodder plants. In addition, the ratio of propionic acid obtained for BTW was similar to the ratios assigned for sainfoin and clover.

## 4. Discussion

The crude protein value (16.51%) of factory black tea waste was between that determined for legume forages (14.34–22.48%). The CP values of legume roughage and BTW were in line with Filya et al. [32]. A roughage’s protein content is a critical factor for microorganism growth [33]. Vegetable hay studied in this experiment, however, had a higher CP content than was required for optimal rumen microorganism activity. Significant differences existed between the feeds in terms of cell wall components (*p* < 0.05). Black tea waste’s ADF and ADL contents were found to be higher than the values determined for legumes, while the NDF content was among the values determined for legumes. NDF values calculated for leguminous feeds were higher than those stated by Canbolat et al. [34]. The ADF and ADL content of legumes agreed with the findings of Filya et al. [31] and Canbolat et al. [34].

This can be explained by the fact that BTW is rich in ADF, ADL, and crude cellulose, from which microorganisms can benefit less. Moreover, GP and energy consumption are positively correlated [35]. In comparison to Filya et al. [32] and Canbolat et al. [34], BTW produced less gas and methane. A notable characteristic of tannins is that they block microbial enzyme activity due to their inhibitory enzyme activities [36]. Consequently, tannin compounds in tea waste may reduce the production of CH_4_ during treatment, which is a desirable effect.

According to the present study, while high energy and protein content positively affected microbial protein production, increased cell wall components negatively influenced it. A similar result was obtained by Canbolat et al. [34].

Metabolic energy contents of legume-dried herbs were found to be in agreement with previous research findings [34,37]. The increase in the gas production value at the 24th hour of fermentation and the crude protein content of the feeds enhanced the OMD value. Additionally, NDF and ADF, which are hard to dissolve in the rumen, limit microbial fermentation, resulting in reduced OMD. OMD values obtained from the present study were in agreement with the values reported by Kamalak et al. [35].

Ruminant rations are restricted by low digestibility and voluntary intake, since ruminants’ voluntary intake is influenced by the cell wall content and digestibility of roughages [34,38]. Alfalfa legume roughage had the lowest NDF, ADF, and ADL content and the highest IVTOMD ratio, while BTW showed the greatest cell wall components and lowest IVTOMD (in vitro true organic matter digestibility). Compared to previous studies, the present study found higher IVTOMD values [34,35,39]. It may be that the differences in roughages used as research materials in the studies explained the results obtained from the present study being greater than the values stated by the researchers.

As a result, an RFV of 100 is calculated for full-bloom alfalfa hay containing 41% ADF and 53% NDF [40]. Differences between feeds were found to be significant in relative feed values (*p* < 0.01). An increase in the components of the cell wall (NDF, ADF, and ADL), which makes digestion of feed difficult, adversely affects RFV. When RFV detected in legume roughage is compared to alfalfa in full bloom, it is seen that alfalfa and clover roughages are second quality, and sainfoin and BTW are third quality. The RFV value determined in legume roughage was found to be lower than that obtained by Adesogan et al. [41] and Canbolat et al. [34].

Compared to Palangi and Macit [16], the current study reported higher acetic acid ratios for legume feeds, and lower propionic acid and butyric acid ratios. In addition, while the acetic acid ratio calculated for black tea waste was similar to the findings reported by Palangi and Macit [16] for some legumes, the propionic acid and butyric acid ratios were found to be lower than in the reports of the researchers in question. A number of factors may have a bearing on the differences among the parameters mentioned in this work and those in previous studies, including animal, method, climate, soil structure, fertilization, plant spice and type, method of application, harvesting time, and vegetation period of feeds. Rumen fermentation and microbial activity can be indicated by changes in VFA profiles. The increase in acetate and reduction in propionate may only indicate an increase in hydrogen accumulation, not a change in methane production. During propionogenesis, hydrogen has not been fully incorporated because propionate has been reduced. That is the case because propiogenesis is a possible alternative receptor that can deliver hydrogen [42]. Consequently, since no increase in methane was observed in these treatments, it is possible that the secondary compounds in tea waste have restricted methanogen access to experimental feeds.

Based on Menke and Steingass [20], the gas produced is only affected by feed properties and chemical composition. According to the results of the present study, the constant ingredients of the tested feed, and the reduction in gas production by adding black tea waste extracts, it is likely that this reduction is a consequence of phenolic compounds’ effect on ruminal microbial populations.

In ruminants, increased concentrations of condensed tannins stimulate protein passage through the rumen, which is probably due to reduced degradation rates by rumen microorganisms as well as decreased growth rates of proteolytic bacteria. Ruminant diets containing condensed tannins may also impair nutrient digestion [43,44] and fermentation [45]. It has been reported that livestock performance can be improved through the proper intake of tannin-containing compounds [46,47]. In light of this, it is critical to determine the appropriate dosage of this phenolic compound without adverse effects on digestion.

Tannins, one of the phenolic compounds found in BTW, can form hydrogen bonds with proteins, which are relatively stable within a pH range of 3.5–8, which prevents their ruminal degradation and reduces the rumen’s ability to degrade nutrients. Once the feed reaches the abomasum, the pH drops below 3.5, breaking these bonds and making the compounds digestible. A tannin prevents bacteria from attaching to plant cell walls, which is crucial for the degradation process [48]. Additionally, complexes formed with proteins and carbohydrates restrict microorganisms’ access to nutrients [49]. In addition to chelating agents, tannins also reduce the availability of metal ions, which rumen microorganisms need for metabolism [50]. Thus, it can be explained why tannin compounds in tea waste resulted in a reduction in methane production in the experiments. 

## 5. Conclusions

It is desirable to utilize waste streams and aim for circularity; in this respect, the utilization of agro-industrial by-products to gain new feedstuff can reduce pollution, for instance, by avoiding uncontrolled disposal and CH_4_ formation into the open atmosphere. Bacteria in fermentation require carbohydrates and protein in order to reproduce, so using these by-products helps preserve the environment and reduce production costs. The results obtained from the current work showed that black tea waste (BTW) has the potential to be used in ruminant rations together with quality roughage. Due to the tannin content of BTW, both the emission of methane gas and the energy loss in the form of methane gas from ruminants may be partially reduced by using it in ruminant rations. Black tea waste extracts could be used to increase feed use efficiency by limiting escaping nutrients from the rumen and preventing the ruminal breakdown of nutrients based on nutritional strategies. It is also necessary to conduct in vivo experiments to find out more about BTW. A large effect of tea waste addition was observed on gas volume, kinetic parameters, and methane production. An increase in methanotrophs is positively correlated with a reduction in methane production. Methane emissions can be reduced by microorganisms, especially those in the methanotroph group, but further research is needed to establish their role. This work has proven the potential of using black tea waste for economic and ecological benefits.

## Figures and Tables

**Table 1 metabolites-13-00731-t001:** Chemical compositions of feeds, (%).

Feed	DM	CA	CP	EE	NDF	ADF	ADL
BTW	92.19	4.59 ^d^	16.51 ^b^	1.35 ^b^	60.21 ^ab^	43.29 ^a^	26.98 ^a^
Sainfoin	92.05	7.24 ^b^	15.97 ^bc^	1.86 ^b^	63.99 ^a^	35.67 ^b^	15.99 ^b^
Clover	92.16	6.86 ^c^	14.34 ^c^	1.98 ^ab^	58.60 ^b^	32.57 ^c^	14.27 ^b^
Alfalfa	91.96	14.77 ^a^	22.48 ^a^	2.73 ^a^	48.77 ^c^	21.64 ^d^	8.87 ^c^
SEM	0.11	0.05	0.60	0.24	1.19	0.37	0.72
*p*	0.450	<0.001	<0.001	0.020	<0.001	<0.001	<0.001

a–d: means with different superscripts within same column are significantly different (*p* < 0.05). SEM: standard error of mean; DM: dry matter (%); CA: crude ash (%); CP: crude protein (%); EE: ether extract (%); NDF: neutral detergent fiber (%); ADF: acid detergent fiber (%); and ADL: acid detergent lignin (%).

**Table 2 metabolites-13-00731-t002:** Measurement values for gas and CH_4_ production (mL/200 mg DM), methane ratios of feeds.

Feed	GP (mL)	Methane (mL)	Methane (%)
BTW	28.44 ^c^	3.44 ^c^	12.09 ^b^
Sainfoin	59.04 ^a^	8.46 ^a^	14.34 ^a^
Clover	54.00 ^a^	8.03 ^ab^	14.89 ^a^
Alfalfa	42.84 ^b^	6.88 ^b^	15.95 ^a^
SEM	2.15	0.44	0.52
*p*	<0.001	<0.001	0.005

a–c: means with different superscripts within the same column are significantly different (*p* < 0.05). SEM: standard error of mean; GP: gas production.

**Table 3 metabolites-13-00731-t003:** Means of MP and performance parameters of feeds.

Feeds	OMD (%)	ME (MJ/kg DM)	NE_L_ (MJ/kg DM)	MP (mg/mL)	IVTOMD (%)
BTW	40.71 ^c^	7.01 ^c^	3.87 ^c^	108.49 ^b^	46.87 ^c^
Sainfoin	66.72 ^a^	11.13 ^a^	6.99 ^a^	118.17 ^b^	57.83 ^b^
Clover	62.34 ^a^	10.36 ^a^	6.41 ^ab^	154.03 ^a^	54.65 ^b^
Alfalfa	53.93 ^b^	9.32 ^b^	5.78 ^b^	115.10 ^b^	62.93 ^a^
SEM	1.82	0.29	0.22	4.37	1.30
*p*	<0.001	<0.001	<0.001	<0.001	<0.001

a–c: means within column with unlike superscript differ significantly (*p* < 0.05). SEM: standard error of means; MP: microbial protein production; OMD: organic matter digestibility; ME: metabolic energy; NE_L_: net energy lactation; and IVTOMD: in vitro true organic matter digestibility.

**Table 4 metabolites-13-00731-t004:** DMD, DMI, and RFV values of feeds.

Feeds	DMD (%)	DMI	RFV (%)	FC
BTW	55.18 ^d^	2.36 ^a^	101.10 ^bc^	3
Sainfoin	61.11 ^c^	2.02 ^b^	95.58 ^c^	3
Clover	63.53 ^b^	2.20 ^ab^	108.22 ^ab^	2
Alfalfa	72.04 ^a^	2.12 ^b^	118.61 ^a^	2
SEM	0.29	0.06	3.29	
*p*	<0.001	0.024	0.006	

a–d: means within column with unlike superscript differ significantly (*p* < 0.05). SEM: standard error of means; DMD: dry matter digestibility; DMI: dry matter intake; RFV: relative feed value; and FC: forage classification.

**Table 5 metabolites-13-00731-t005:** Average values of volatile fatty acids composition of feeds.

Feed	AA mmol/L	PA mmol/L	BA mmol/L	AA %	PA %	BA %	TVFAmmol/lt
BTW	46.82 ^c^	10.85 ^d^	6.96 ^c^	69.58 ^a^	16.12 ^b^	10.34 ^b^	67.29 ^c^
Sainfoin	52.36 ^b^	13.46 ^c^	10.37 ^b^	65.71 ^b^	16.87 ^b^	13.01 ^a^	79.71 ^b^
Clover	57.00 ^a^	15.06 ^b^	12.43 ^a^	65.23 ^b^	17.24 ^b^	14.22 ^a^	87.38 ^a^
Alfalfa	56.60 ^a^	17.20 ^a^	9.79 ^b^	63.56 ^c^	19.31 ^a^	11.00 ^b^	89.05 ^a^
SEM	0.58	0.43	0.40	0.40	0.41	0.46	0.92
*p*	<0.001	<0.001	<0.001	<0.001	0.003	0.001	<0.001

a–d: means within column with unlike superscript differ significantly (*p* < 0.05). SEM: standard error of means; AA: acetic acid; PA: propionic acid; BA: butyric acid; TVFA: total volatile fatty acids.

## Data Availability

No new data were created or analyzed in this study. Data sharing is not applicable to this article.

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
