# Peer review of "Comparison of Black Tea Waste and Legume Roughages: Methane Mitigation and Rumen Fermentation Parameters"

_metabolites, 2023, doi:10.3390/metabo13060731_

Round 1

Reviewer 1 Report

Dear Authors I have revised your manuscript, but to consider this paper I need to have more informations about the experiment, because methods are not completely described. You do not write who many samples you analyzed, you used acronyms not described. You put in the tables parameters not described in materials and methods.

So I suggest to describe better the experiment.

Than I can understand if there are the conditions to be published and I will give you indications to revise the paper in each parts.

Best Regards

Author Response

Dear referee, the materials and methods section was revised.

Best regards, Maximilian Lackner

Reviewer 2 Report

Very poor manuscript which has major short comings in the Material and Method section and therefore does not allow to repeat the study. Calculation of values is not properly explained. Some analysed values look rather strange or may be even wrong. The discussion repeats in numerous cases the results and the results were only related to the tannin content which have not been measured in all samples.

Line 103-114 no information of diet of donor animal is presented

Line 120 according to AOAC there are several methods given each oft proximate analysis (DM, CP, etc). Therefore it would be useful if you give the method No of each analysis

Line 122- 123 We need more information here! How much sample was used for inoculation? At which temperature did you inoculate? In what kind of glass ware was fermentation performed? How can you read the gas production in ml? How many blanks did you use to correct for gasprouction of rumen fluid (there s still some fermentable rest in rumen fluid)? Did you use any standards for corrections etc? The description of gasproduction method does not allow repetition. Please include the number of observations per feedstuff. How many runs (repetition) per feedstuffs on any other day were  performed?

Line 125 How did you calibrate the Sensor??

Line 129 These are not performance values…Feed is not performing, only the animals perform …Please  indicate precisely which formulas you used for calculations of these feed values ME , Nel, OMD.. The best would be to indicate the formulas in the text

Line 131-133 Please indicate which  formula you used  for calculation of DMI, DMD and RFV etc . there are several formuas in the reference …please include in the text.

Line 135-140 Please include formulas for calculation of the values.

The authors do not clearly indicate how there values were calculated , one cannot expect that the reader starts searching which formula you took for calculation!! Please insert a section calculations and include all formulas you used.

Line 140-143 PLease include a short description of VFA Analysis and apparatus used

Table 1. How can you do statistics on feed samples? This is n=1!! According to your description you have only  one sample and not several samples of BTW and legumes. Multiple analysis of the same sample gives only the variation of the analysis and not between feed samples. Please delete statistics from table 1 and just compare your results without statistics.

Table 1 the values of ADL for sainfoin, clover and BTW are probably not correct..they are much too high!!

Line 159-160 condensed tannin content for BTW is given. Please include method of analysis in Materia and Method section. Where ist he condensed tannin content  of the legumes?? How was rumen fluid ammonia measured? Give method in Material and Method section.  The abbreviation for liter is l…and not lt… what do you mean with 3.02, 65.75 and 150.82 %??

before it is mg/l

see line 156, 168 and elswhere… do not use significant and P-value… one is sufficent to use

line 169 the ranking should : sainfoin=clover >alfalfa>BTW…  there is no significant difference between sainfoin and clover

Table 3 Microbial protein  is given in mg/ml …what does that mean ? is that per ml inoculum, but then you have in in two ml already more protein than sample in in 30 ml inoculum

Table 3 ME are given per kg KM what is KM??.  Megajoule is abrreviated with capital letters  MJ

Table 3 this is irritating, true digestibility values are higher compared to OMD…PLease explain what happened to sainfoin and clover IVTODM values???

Line 184 metabolizable energy

Line 185 dried grass?There were no grass samples in this study!!!

Line 186 -190  this is no description of results…this belongs into the discussion…

Line 188 which crude protein rate do you mean??   You only measured crude protein content…gasproduction is correlated with carbohydrate concentration…the low energy content in alfalfa is probably related to the CA value

Table 4? What is here shown? DMI in % ? for which animals ist hat in% bodyweight or what what do you mean.  Relative feed value …what does this measurement mean..this needs tob e explained.

Line 222—234 this is mainly repetition of the results …this is not needed

line 241 the legumes contain tannins as well…it would be good to measure tannin content of all samples for comparison

line 262 usually the ADF content is lower than NDF, because ADF is a part of NDF…something is wrong here

line 263- 267 again repetition of results …should be avoided

line 270 -286  I do not know what the authors want to say with this paragraph … no idea what they are comparing here… you need to include values from other studies so that the reader see differences that you want to explain

line 300- 309 all legumes contain tannins what do you want to explain

line 303 after the abomasum , the tannins reach the intestine which has a pH of 6.5 and tannins will rebuild bindings with protein and decrease intestinal protein digestion that has been shown in numerous studies

The reduction of methane is mainly by reduced fiber degradabilty leaving less energy for the animal and reduces the performance when included in the diet...this does not help to reduce methane emission!

Author Response

Dear reviewer, see attached our comments. We have addressed all points. Thanks for your support!

Best regards, Maximilian Lackner

Reviewer 3 Report

The title as well as keywords accurately reflects the major findings of the work.

The abstract adequately summarize methodology, results, and significance of the study. However, Authors should better describe the statistical analysis applied on data.

The introduction section is well written and it falls within the topic of the study. However, more information regarding livestock nutrition and the usefulness of supplementary diet on livestock production should be added in order to make stronger the rationale of the Authors. At this regards, at line 43, after the sentence “Several countries use alternative feed sources to supplement ruminant animals' nutrition caused by a lack of quality roughage.” I suggest to add “As a matter of facts, in recent years, nutritional strategies have emerged and it has been proposed as a key factor to improve the health status and welfare of animals (Abbate, J.M. et al. Animals, 2020, 10(12), pp. 1–13, 2303) as well as to enhance productivity in livestock (Armato L. et al., Acta Agriculturae Scandinavica, Section A, 2016, 66: 119-124; Monteverde V. et al., Journal of Applied Animal Research, 2017, 45: 615-618). Actually, the main compounds used to this purpose are fermented feed, probiotics, prebiotics, synbiotic or various types of residues from plant production.”

The section of Materials and Methods is clear for the reader and it meticulously describes the methods applied in the study.  However, Authors should check this section and correct many punctuation errors. Regarding statistical analysis, Did Authors perform a normality test on data in order to assess their normal distribution? Please clarify this aspect and specify whether data passed normality test before parametric analysis application.

Results section as well as Discussion section is clear and well written.

The findings obtained in the study were well discussed and justified with appropriate references. The conclusion section is well written; Authors well summarize the results and emphasize the significance of the study appropriately. 

Tables are generally good and well represent the results of the study.

Author Response

I am grateful for the valuable comments of the esteemed referee. Yes, the data were normalized before analysis and outliers were removed.

Best regards, Maximilian Lackner

Reviewer 4 Report

Dear Authors,

I revised the manuscript Comparison of Black Tea Waste and Some Legume Roughages: Methane Mitigation and Rumen Fermentation Parameters” submitted to the Metabolites Journal. The authors of the article raise a very important problem of reducing methane emissions during cattle breeding. Thanks to the study, you can learn more about the use of waste from the production of black tea in feeding ruminants. The article is interesting. However, I do have some concerns that should be noted.

 1. Introduction

Line 42-43. Please provide these alternative feed sources and the countries where these feeds are used.

Line. 77-78. Please write what percentage of methane emissions are responsible for animals, including ruminants.

3. Results

3.1. Chemical composition

Line 159-160. In what units is the content of tannins given? Correct the units in the text. Why is there a % after the number 150.82.

3.3. MP, OMD, ME and NEL levels of feeds

Line 176. Please use the correct unit for ME and NEL, i.e. "MJ" and not "mj" or possibly after converting "kcal". What does "KM" mean in these units should not be DM?

3.5.    The effect of feeds on rumen volatile fatty acid levels

Line 210. What does "lt: in mmol/lt" mean in the table?

Author Response

(The authors gave the same response as above.)

Reviewer 5 Report

44 waste - do you mean BTW?

111 what do you mean rumen development was she a cow or a heifer- be specific?

117 BTW be consistent with abbreviations throughout the paper.

131 How was DMI measured?

137-138 please correct what is TOMD?

139 define OM

144 what was the experimental design

156 Omit significant add the P value after differences

159 160 mg/l

180-194 BTW decreased methane, but also decreased OMD

230 here and throughout do not begin a sentence with an abbreviation

279 microbial activity

300 Use abbreviations correctly 

btw

308 ...need for  metabolism

Author Response

(The authors gave the same response as above.)

Round 2

Reviewer 1 Report

Thank you  for your revisions

Reviewer 2 Report

The authors included some half hearted corrections but were not able to improve the critical points of the manuscript

110 …Brown Swiss castles  ?

111 it is sufficient to write ..euthanized…I don’t believe that the animals feels that it was humanely  or do you mean it was human for the person who slaughtered the animal...

113 We know what is recommended, we want to know what you really did!!

114 mixture rations..several rations?

114-115 what is a maintenance ratio…do you mean maintenance requirement???

122 948.13 is the method for Glycerol In eggs???

123 do you mean 942.05?

131 why do you need 3 blanks per each treatment, blanks only contain rumen fluid they are independent from treatment

131/132 please give full supplier details for standards

136 – 139 In th epaper by Menke and Steingass there are several equations given for calculation of ME, and OM and NEL …please include the precise equation you used …The way it is preented it is not retraceable for the reader what you calculated

143- 148  Please give the precise regression equation you used for calculation OF DDMI; DDM;    based on ADF and NDF content …

Line 149 in reference 24, 25 and 26 and as well  27 is no description how to calculate/analyse protein biomass from rumen bacteria. Reference28 is written in turkish and probably not understandable for many readers. It is not retraceable form e what has been calculated here. This needs improvement

Line 155- 156 It is now even more complicated. Where do this bags come from. Hohenheim gas production has no bags. Please give more information here. This is not understandable for the reader

Line 166-171 reference for tannin analysis is missing…Did you measure total tannins or only condensed tannins please explain

Table 1: There is only one feed sample for each plant material according to Material and Methods, so in Table 1 statistics is done on n=1 samples, please delete statistics from table 1 and corresponding text. To analysis three timest he sample gives only the standard error of the analysis and not between feeds. Table 1 and P value is never zero.

Line 189 There is only tannin content of BTW, where is the tannin content of sainfoin, clover and alfalfa??? What is unit oft he tannin content line 189 (3.02???)

Table 3 the problem that iVtrue OM digestibility is lower than OMD has not been fixed. True digestibility is always higher and  is corrected for endogenous losses how do you measure endogenous losses for invitro??

Line 211 and Table 3 there is no retraceable explanation how MP was calculated and the references given d o not indicate how this was measured

Line 293 full bloom alfalfa cannot contain 41% NDF and 53% ADF…ADF is always lower than NDF

The discussion has not been changed at all .

Author Response

The authors included some half hearted corrections but were not able to improve the critical points of the manuscript

110 …Brown Swiss castles  ?

Thanks for your comment. It was corrected.

111 it is sufficient to write ..euthanized…I don’t believe that the animals feels that it was humanely  or do you mean it was human for the person who slaughtered the animal...

I didn't use the word " slaughter " because the slaughterhouse followed the ethics of killing animals and they were killed with minimal pain.

113 We know what is recommended, we want to know what you really did!!

Thanks for your comment. It was corrected.

114 mixture rations..several rations?

Thanks for your comment. Mixed ration is correct.

114-115 what is a maintenance ratio…do you mean maintenance requirement???

This was to maintain relatively stable rumen conditions. It was added.

122 948.13 is the method for Glycerol In eggs???

Thanks for your comment. It was corrected to 976.05.

123 do you mean 942.05?

Thanks for your comment. It was corrected.

131 why do you need 3 blanks per each treatment, blanks only contain rumen fluid they are independent from treatment

In order to correct the produced gas accordingly. In other words, reduce the gas produced by rumen microorganisms.

131/132 please give full supplier details for standards

Hohenheim standard alfalfa has been presented in the manuscript. “Based on standard (University of Hohenheim) measurements, the produced gas was measured after correction with 49.61 ml per 0.200 g dry matter.”

136 – 139 In th epaper by Menke and Steingass there are several equations given for calculation of ME, and OM and NEL …please include the precise equation you used …The way it is preented it is not retraceable for the reader what you calculated

Thanks for your comment. It was added.

143- 148  Please give the precise regression equation you used for calculation OF DDMI; DDM;    based on ADF and NDF content …

Thanks for your comment. It equation was presented previously.

Line 149 in reference 24, 25 and 26 and as well  27 is no description how to calculate/analyse protein biomass from rumen bacteria. Reference28 is written in turkish and probably not understandable for many readers. It is not retraceable form e what has been calculated here. This needs improvement

Thanks for your comment. This book is for analyses methods.

Line 155- 156 It is now even more complicated. Where do this bags come from. Hohenheim gas production has no bags. Please give more information here. This is not understandable for the reader

Thanks for your comment. The bags were for OM experiment, its not for gas production.

Line 166-171 reference for tannin analysis is missing…Did you measure total tannins or only condensed tannins please explain

 Thanks for your comment. It was added.

Table 1: There is only one feed sample for each plant material according to Material and Methods, so in Table 1 statistics is done on n=1 samples, please delete statistics from table 1 and corresponding text. To analysis three timest he sample gives only the standard error of the analysis and not between feeds. Table 1 and P value is never zero.

 Thanks for your comment. This is not a scientific conclusion, it shows the dispersion of p criterion in statistics, which probably, the dispersion of our data was close to the average, and the p criterion is zero. We had 3 samples in each replicate and our analysis resoults are absolutly correct.

Line 189 There is only tannin content of BTW, where is the tannin content of sainfoin, clover and alfalfa??? What is unit oft he tannin content line 189 (3.02???)

Thanks for your comment. The main purpose of the experiment was to investigate the possibility of using BTW as an alternative feed, so we focused on it for this reason.

Table 3 the problem that iVtrue OM digestibility is lower than OMD has not been fixed. True digestibility is always higher and  is corrected for endogenous losses how do you measure endogenous losses for invitro??

Thanks for your comment. OMD and TOMD calculation formols are different.

Line 211 and Table 3 there is no retraceable explanation how MP was calculated and the references given d o not indicate how this was measured

Thanks for your comment. It was added.

Line 293 full bloom alfalfa cannot contain 41% NDF and 53% ADF…ADF is always lower than NDF

The discussion has not been changed at all .

Thanks for your comment. It was corrected.

Reviewer 4 Report

Dear Authors,

I have revisited the manuscript “Comparison of Black Tea Waste and Some Legume Roughages: Methane Mitigation and Rumen Fermentation Parameters” submitted to the Metabolites Journal. The work has been improved according to my recommendations. The article is fully suitable for publication in Metabolites Journal.

Reviewer 5 Report

line 110 cattle not castles.

I wonder if you need to talk about rumen development at al as she was a 7 year old cow.

Author Response

line 110 cattle not castles.

The reviewer’s suggestion was great. It was corrected.

I wonder if you need to talk about rumen development at al as she was a 7 year old cow.

Rumen development was not considered in the above research and we needed rumen fluid to simulate the rumen environment in the laboratory.